# Influence to Hardness of Alternating Sequence of Atomic Layer Deposited Harder Alumina and Softer Tantala Nanolaminates

**Helle-Mai Piirsoo** * , **Taivo Jõgiaas** , **Peeter Ritslaid** , **Kaupo Kukli** and **Aile Tamm**

Institute of Physics, University of Tartu, W. Ostwaldi Str. 1, 50411 Tartu, Estonia; taivo.jogiaas@ut.ee (T.J.); peeter.ritslaid@ut.ee (P.R.); kaupo.kukli@ut.ee (K.K.); aile.tamm@ut.ee (A.T.)
* Correspondence: helle-mai.piirsoo@ut.ee

**Abstract:** Atomic layer deposited amorphous 70 nm thick $Al_2O_3$-$Ta_2O_5$ double- and triple-layered films were investigated with the nanoindentation method. The sequence of the oxides from surface to substrate along with the layer thickness had an influence on the hardness causing rises and declines in hardness along the depth yet did not affect the elastic modulus. Hardness varied from 8 to 11 GPa for the laminates having higher dependence on the structure near the surface than at higher depths. Triple-layered $Al_2O_3/Ta_2O_5/Al_2O_3$ laminate possessed the most even rise of hardness along the depth and possessed the highest hardness out of the laminates (11 GPa at 40 nm). Elastic modulus had steady values along the depth of the films between 145 and 155 GPa.

**Keywords:** nanoindentation; nanolaminates; atomic layer deposition; nanoengineering





## 1. Introduction

Nanolaminates are being investigated for the possibility of nanoengineering, i.e., constructing thin films with a combination of multiple physical properties exactly suitable for a given application. Customizable properties dependent on the layered structure range from mechanical hardness [1–8] to magnetic susceptibility [2,9–11], and refraction properties [1].

The structure of layers constituting nanolaminates influences the mechanical properties of the film. For instance, different order of $ZrO_2$ and $SnO_2$ layers from surface to substrate in only 30 nm thick bilayer thin film caused a change of about 5 GPa in hardness and 100 GPa in the modulus [2]. For TiN/TiBN nanolayered films, it was found, that the layer thickness influenced film adhesion, wear resistance, and frictional coefficient in addition to hardness and elastic modulus [3]. The number of layers in ZnO/graphene nanolaminate influenced the hardness by ±6 GPa and elastic modulus by ±100 GPa [4]. ZrN layer thickness in reactive magnetron sputtered ZrN/SiNx nanolaminates had a correlation with the intrinsic growth stress [5]. The layered structure of ALD $Al_2O_3/TiO_2$ films also has had an influence on the residual stress in the film [6,7], as well as their hardness and elastic modulus [8].

Atomic layer deposition (ALD) of $Al_2O_3$ with $Al(CH_3)_3$ and $H_2O$ precursors is a widely studied and known process [12,13]. $Ta_2O_5$ combined with $Al_2O_3$ in laminate thin films have been proposed for various applications due to their promising electrical properties, often dependent on the layered structure. Amorphous $Al_2O_3$-$Ta_2O_5$ quadruple-layered laminate showed higher dielectric strength compared to bilayers and laminates with eight layers [14]. The operation of high voltage metal-insulator-metal capacitors with $Al_2O_3$-$Ta_2O_5$ bilayers depended on the $Al_2O_3/Ta_2O_5$ thickness ratio [15]. $Al_2O_3$-$Ta_2O_5$ laminates have also exhibited resistive switching characteristics [16] and good corrosion resistance [17]. ALD of $Ta_2O_5$ with $Ta(OEt)_5$ and $H_2O$ precursors has been developed before [18–22]. Deposition of $Al_2O_3$ or $Ta_2O_5$ at 300 °C results in amorphous oxides for both of the aforementioned ALD processes. The mechanical hardness of amorphous atomic layer deposited $Al_2O_3$ thin

films lays between 7 and 12 GPa. It has been reported to be 10.5 [23] and 12 [24] GPa on Si substrates, 11 GPa on TiN/Si [25], 9.5 GPa on soda-lime glass [26], and 7.2 [27] and 11.8 [28] GPa on steel substrates. The elastic modulus of these $Al_2O_3$ films has varied from 68 (on glass) to 260 (on steel) GPa, while samples with Si substrate possessed intermediate values (145, 150, 220 GPa) [23–26,28]. The mechanical hardness of atomic layer deposited $Ta_2O_5$ film on a soda-lime glass substrate has been found to be 6.7 GPa, while the modulus was 96 GPa [26]. The hardness has been measured to be lower than that for $Ta_2O_5$ thin films fabricated with other methods—4.7 GPa for e-beam deposition on Co-Cr substrate [29] and 5.2 GPa for physical vapor deposition on Ti-6Al-4V substrate [30]. However, the elastic modulus was slightly higher in the films obtained with other methods compared to that in the films grown by ALD, i.e., 160 GPa with ion beam sputtering on Si [31] and 119 GPa with e-beam deposition on Co-Cr substrate [29].

In this work, double- and triple-layered amorphous $Al_2O_3$-$Ta_2O_5$ laminates with an overall thickness of about 70 nm were atomic layer deposited while changing the sequence of the layers from surface to substrate. Hardness and elastic modulus of the laminates were measured with nanoindentation, and the effects of the constituent layer thickness and deposition sequence on the mechanical properties of the laminates were analyzed.

## 2. Materials and Methods

### 2.1. Film Deposition

The layered films were produced by atomic layer deposition (ALD) in an in-house built reactor [32]. The metal precursors were $Al(CH_3)_3$ and $Ta(OC_2H_5)_5$, while $H_2O$ was used as the oxygen source. Pulse times were 2/2/2/5 s for metal precursor, $N_2$ purge, water and the second purge, respectively. Evaporation temperature for $Ta(OC_2H_5)_5$ was $95 \pm 5$ °C, and the growth temperature for both $Al_2O_3$ and $Ta_2O_5$ was $300 \pm 10$ °C. Si (100) wafers were exploited as substrates. The number and sequence of deposition cycles are presented in Table 1. Reference $Al_2O_3$ and $Ta_2O_5$ films were deposited along with two double-layer laminates and two triple-layer laminates with different layering orders (Figure 1). An additional quadruple-layered laminate was also prepared.

**Table 1.** Denotations of films with the number of atomic layer deposition cycles with $Al_2O_3$ representing the $Al(CH_3)_3$ and water cycle and $Ta_2O_5$ representing the $Ta(OC_2H_5)_5$ and water cycle. Layer thicknesses according to X-ray reflectometry are presented in order from substrate to surface.

| Denotation of Film | No. of ALD Cycles | Layer Thickness (nm) |
|---|---|---|
| $Al_2O_3$/Si | $600 \times Al_2O_3$ | 69 nm $Al_2O_3$ |
| $Ta_2O_5$/Si | $800 \times Ta_2O_5$ | 61 nm $Ta_2O_5$ |
| $Ta_2O_5$/$Al_2O_3$/Si | $340 \times Al_2O_3 + 470 \times Ta_2O_5$ | 38 nm $Al_2O_3$ + 35 nm $Ta_2O_5$ |
| $Al_2O_3$/$Ta_2O_5$/Si | $470 \times Ta_2O_5 + 340 \times Al_2O_3$ | 34 nm $Ta_2O_5$ + 40 nm $Al_2O_3$ |
| $Al_2O_3$/$Ta_2O_5$/$Al_2O_3$/Si | $227 \times Al_2O_3 + 313 \times Ta_2O_5 + 227 \times Al_2O_3$ | 29 nm $Al_2O_3$ + 23 nm $Ta_2O_5$ + 27 nm $Al_2O_3$ |
| $Ta_2O_5$/$Al_2O_3$/$Ta_2O_5$/Si | $313 \times Ta_2O_5 + 227 \times Al_2O_3 + 313 \times Ta_2O_5$ | 23 nm $Ta_2O_5$ + 24 nm $Al_2O_3$ + 24 nm $Ta_2O_5$ |
| $Ta_2O_5$/$Al_2O_3$/$Ta_2O_5$/$Al_2O_3$/Si | $170 \times Al_2O_3 + 268 \times Ta_2O_5 + 170 \times Al_2O_3$ $+ 268 \times Ta_2O_5$ | 19 nm $Al_2O_3$ + 19 nm $Ta_2O_5$ + 19 nm $Al_2O_3$ $+ 20$ nm $Ta_2O_5$ |

**Figure 1.** Schematics illustrating the layered structure of the laminates as the notation for each of the laminates is shown above.

## 2.2. Film Characterization

The elemental composition of the films was determined by wavelength dispersive X-ray fluorescence (WD-XRF) spectroscope ZSX-400 (Rigaku, Tokyo, Japan). The surface of the films was characterized by scanning electron microscopy (SEM) using Helios NanoLab 600 (FEI, Hillsboro, OR, USA). The phase composition of the films was determined by grazing incidence X-ray diffraction (GIXRD). The layered structure of the films was confirmed with the X-ray reflectivity (XRR) method. The XRD and XRR measurements were done with SmartLab$^{TM}$ (Rigaku, Tokyo, Japan) diffractometer, and for the analysis, AXES [33] software was used. In GIXRD measurements, grazing angle was 0.5°, scan step 0.04°, and scan rate 3°/min. Grazing angle varied till 7°, scan step and rate were 0.01° and 1.5°/min, respectively, for the XRR measurements.

## 2.3. Nanoindentation

Nanoindentation was carried out with Hysitron TriboIndenter TI980 (Bruker, Billerica, MA, USA) with a Berkovich tip. Peak load of 0.5 mN was applied in continuous stiffness measurement mode with tip frequency of 220 Hz, which obtains several tens of data points over a displacement range for a single indentation [34]. Loading, peak hold, and unloading times were set, respectively, to 35, 2, and 5 s. Fifteen indents in total were laterally separated by 10 μm. Gathered data were analyzed (linear interpolation, averaging) with Origin 2020 software. Prior to the measurements tip was calibrated on a fused quartz glass with a reduced modulus of 69.6 GPa and hardness of 9.25 GPa (Figure 2). The TriboIndenter was used in scanning probe microscopy mode to image the indents.

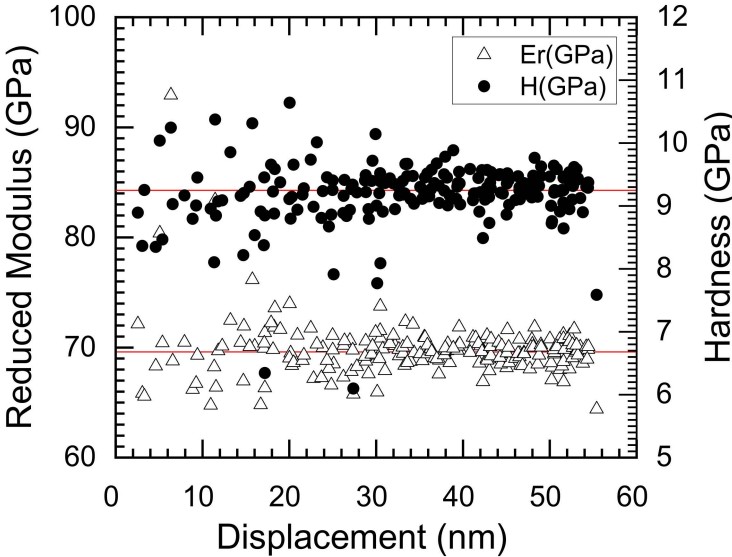

**Figure 2.** Calibration measurements for reduced modulus (▲) and hardness (●) along the depth of the reference quartz sample.

## 3. Results

### 3.1. Elemental Composition and Microstructure

The ALD growth rates were 0.11 nm/cycle for $Al_2O_3$ and 0.07 nm/cycle for $Ta_2O_5$ as determined with XRR and XRF methods and confirmed with ellipsometry (not shown) for reference samples. XRF results are presented in Table 2, revealing atomic ratios appreciably close to the stoichiometric $Al_2O_3$ and $Ta_2O_5$, as expected for the ALD processes [13,22].

SEM measurements revealed quite smooth, i.e., featureless surfaces for all the deposited films. An example surface is shown in Figure 3a. All the deposited films were X-ray amorphous, exemplified by the diffractogram of the laminates in Figure 3b. XRR measurement fittings confirmed the layered structures and thicknesses are presented in

**Table 1.** Average density of the oxides was 3.1 ± 0.2 g/cm$^3$ and 8.4 ± 0.2 g/cm$^3$ for Al$_2$O$_3$ and Ta$_2$O$_5$ layers, respectively.

**Table 2.** Elemental composition of the ≈70 nm thick films according to XRF.

| Film | Al (at.%) | Ta (at.%) | O (at.%) |
|---|---|---|---|
| Al$_2$O$_3$/Si | 41 | - | 59 |
| Ta$_2$O$_5$/Si | - | 25 | 75 |
| Ta$_2$O$_5$/Al$_2$O$_3$/Si | 21 | 13 | 66 |
| Al$_2$O$_3$/Ta$_2$O$_5$/Si | 20 | 12 | 68 |
| Al$_2$O$_3$/Ta$_2$O$_5$/Al$_2$O$_3$/Si | 27 | 8 | 65 |
| Ta$_2$O$_5$/Al$_2$O$_3$/Ta$_2$O$_5$/Si | 13 | 17 | 70 |
| Ta$_2$O$_5$/Al$_2$O$_3$/Ta$_2$O$_5$/Al$_2$O$_3$/Si | 19 | 14 | 67 |

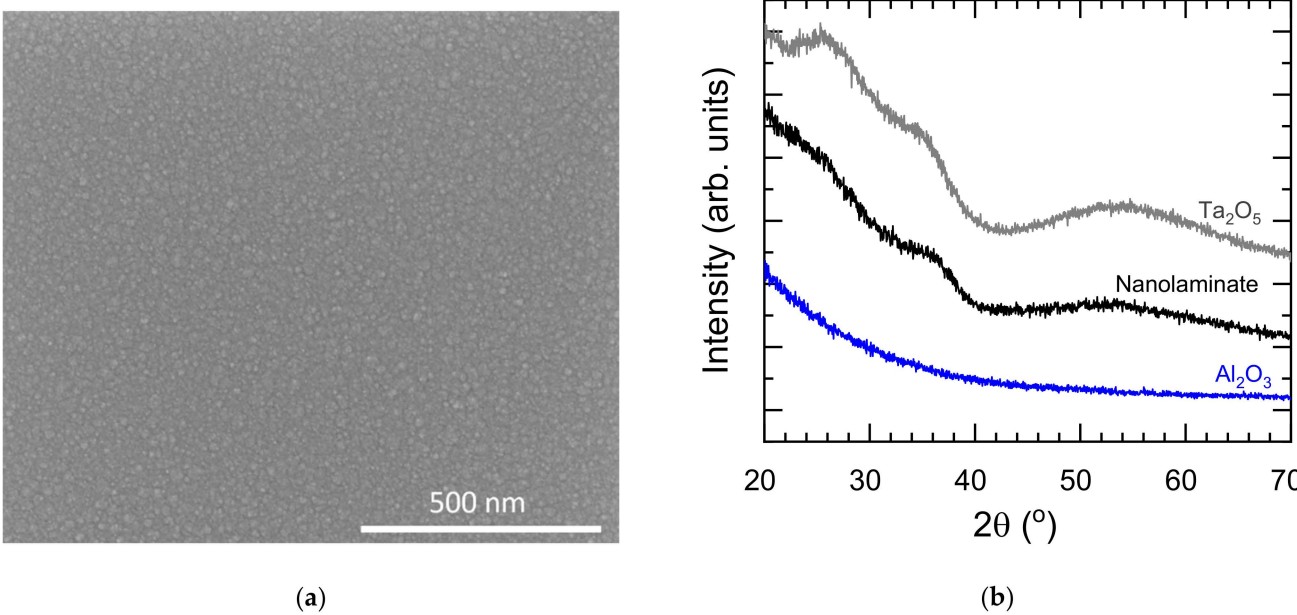

(**a**)　　　　　　　　　　　　　　　　　　　　　　　　　(**b**)

**Figure 3.** (**a**) SEM image of the surface of as-deposited Al$_2$O$_3$/Ta$_2$O$_5$/Al$_2$O$_3$/Si laminate, (**b**) GIXRD diffractograms of Al$_2$O$_3$-Ta$_2$O$_5$ nanolaminate and reference Ta$_2$O$_5$ and Al$_2$O$_3$ films.

*3.2. Mechanical Properties of Amorphous Laminates*

　　Figure 4 depicts the averaged hardness values with standard deviation for reference samples: Al$_2$O$_3$/Si, Ta$_2$O$_5$/Si, and the four-layered Ta$_2$O$_5$/Al$_2$O$_3$/Ta$_2$O$_5$/Al$_2$O$_3$/Si laminate films. The four-layered laminate is considered as a reference because it possessed a homogeneous hardness across displacements in the approximate range of the film thickness. The Si substrate possessed an average hardness of 13.5 ± 0.4 GPa and an average elastic modulus of 147 ± 3 GPa (not shown).

　　The hardness of Al$_2$O$_3$ had a steady incline with depth from around 12 GPa, after displacement of 11 nm, to 13 GPa at 40 nm. Similarly, the hardness of Ta$_2$O$_5$ film rose from 8 GPa at 11 nm to 10 GPa at 45 nm. The Al$_2$O$_3$ and Ta$_2$O$_5$ films exhibited a difference in hardness around 3–4 GPa over the measured displacement range. For the laminate, the thickness of layers is presented on the graph with dotted lines and arrows. The hardness of sample Ta$_2$O$_5$/Al$_2$O$_3$/Ta$_2$O$_5$/Al$_2$O$_3$/Si started from 9.5 GPa at 10 nm and then continued with a steady incline to 11 GPa with depth. The hardness-displacement graphs did not indicate the occurrence of indentation size effect, i.e., a rise in hardness near surface, which has been recorded for metal films in a similar displacement range [35].

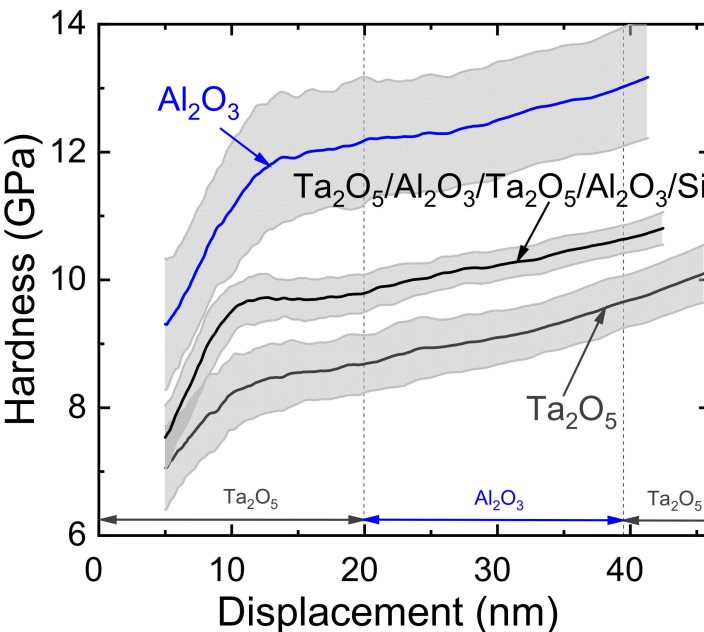

**Figure 4.** Averaged hardness of 15 indents with standard deviation shown as grey areas. Vertical dotted lines and arrows indicate the thickness and sequence of three topmost layers for the $Ta_2O_5/Al_2O_3/Ta_2O_5/Al_2O_3/Si$ laminate.

Figure 5a presents the hardness for the two double-layered laminates (Figure 1). For the $Al_2O_3/Ta_2O_5/Si$ film, the different mechanical hardness of individual layers is apparent decided on the basis of the wavelike shape of the curve as the hardness shows a maximum (10.5 GPa) at a displacement of 11 nm and a minimum (9.9 GPa) at 25 nm of depth. It should be noted that the interface between the layers was formed at 39 nm (Figure 5a).

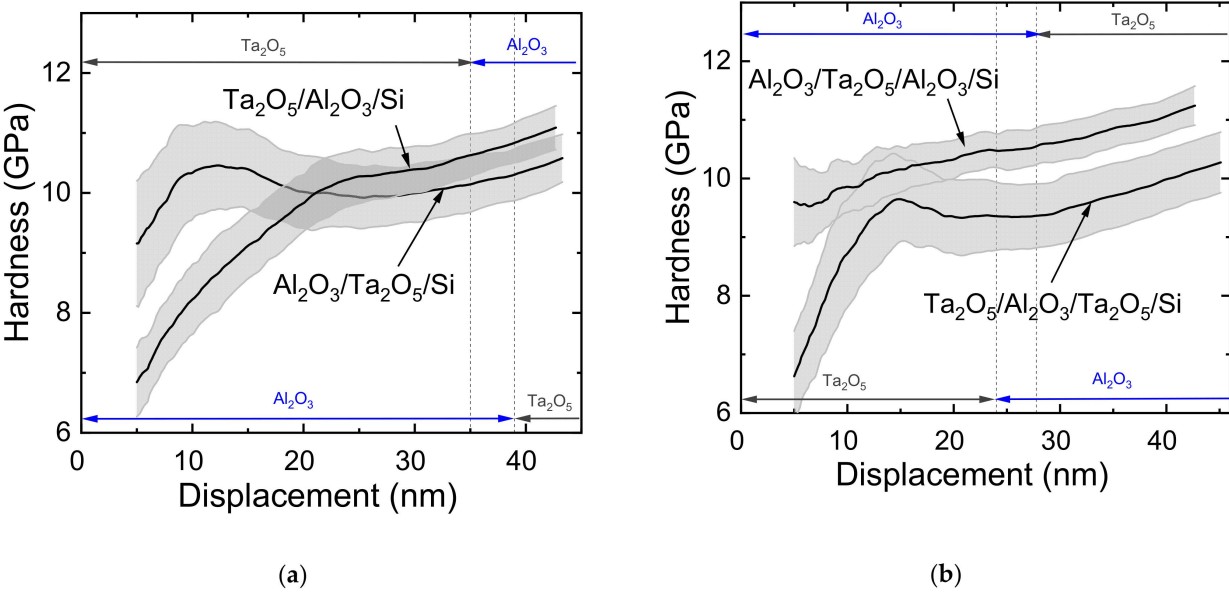

(a)　　　　　　　　　　　　　　　　　　　(b)

**Figure 5.** (**a**) Averaged hardness of 15 indents with standard deviation shown as grey areas. Vertical dotted lines and arrows indicate the thickness and sequence of layers for the double-layered laminates. (**b**) Averaged hardness of 15 indents with standard deviation shown as grey areas. Vertical dotted lines and arrows indicate the thickness and sequence of layers for the triple-layered laminates. Only two of the topmost layers are visualized on the graph.

For the $Ta_2O_5/Al_2O_3/Si$ laminate, the hardness depicted a steep rise until 20 nm of depth, however, the angle of incline changed twice, it decreased at 20 nm and slightly increased again at 33 nm. The interface between $Ta_2O_5$ and $Al_2O_3$ layer was at 35 nm.

Figure 5b depicts the average hardness for triple-layered laminates. The hardness of the $Ta_2O_5/Al_2O_3/Ta_2O_5/Si$ laminate changed similarly along the depth to the wavelike shape of the indentation curve for the laminate $Al_2O_3/Ta_2O_5/Si$. Even though the system, in this case, is three-layered and consists mostly of the softer $Ta_2O_5$, the maximum was still quite high (9.6 GPa) at 15 nm and the minimum (9.3 GPa) at 25 nm. At the same time, $Al_2O_3$ layer begins at 24 nm and ends at 47 nm from the surface.

The second triple-layered laminate possessed a steady incline of hardness along the depth of the film (Figure 5b), which was similar to the behavior of the quadruple-layered laminate.

The summary of the hardness and moduli values of the films at three depths (10, 25, and 40 nm) are shown in Table 3. The surface was softer (8.2–9.5 GPa) for films with $Ta_2O_5$ top layer. The surface was the hardest at 11 GPa for the $Al_2O_3/Si$ film and was lower by about 1 GPa for the $Al_2O_3/Ta_2O_5/Si$ film. At deeper displacements, the hardness values overlapped more significantly, yet the triple-layered laminate $Al_2O_3/Ta_2O_5/Al_2O_3/Si$ possessed slightly higher hardness compared to the rest of the laminates. Double- and quadruple-layered laminates had similar hardness at 25 and 40 nm displacements (10–11 GPa), while the triple-layer laminate with the largest amount of $Ta_2O_5$ was the softest among all the laminates.

**Table 3.** Average hardness and reduced modulus with standard deviation (SD) at various displacements.

| Film | Hardness ± SD at 10 nm | Modulus ± SD at 10 nm | Hardness ± SD at 25 nm | Modulus ± SD at 25 nm | Hardness ± SD at 40 nm | Modulus ± SD at 40 nm |
|---|---|---|---|---|---|---|
| $Al_2O_3/Si$ | 11.2 ± 1.0 | 154 ± 9 | 12.3 ± 0.9 | 160 ± 6 | 13.1 ± 0.9 | 160 ± 6 |
| $Ta_2O_5/Si$ | 8.2 ± 0.6 | 139 ± 5 | 8.9 ± 0.5 | 141 ± 4 | 9.7 ± 0.4 | 139 ± 3 |
| $Ta_2O_5/Al_2O_3/Si$ | 8.2 ± 0.6 | 146 ± 6 | 10.3 ± 0.5 | 155 ± 3 | 10.9 ± 0.4 | 156 ± 2 |
| $Al_2O_3/Ta_2O_5/Si$ | 10.4 ± 0.8 | 145 ± 6 | 9.9 ± 0.5 | 151 ± 4 | 10.4 ± 0.4 | 156 ± 4 |
| $Al_2O_3/Ta_2O_5/Al_2O_3/Si$ | 9.9 ± 0.4 | 145 ± 4 | 10.5 ± 0.3 | 148 ± 3 | 11.1 ± 0.3 | 148 ± 3 |
| $Ta_2O_5/Al_2O_3/Ta_2O_5/Si$ | 8.7 ± 0.9 | 147 ± 5 | 9.4 ± 0.6 | 151 ± 3 | 10.0 ± 0.5 | 153 ± 3 |
| $Ta_2O_5/Al_2O_3/Ta_2O_5/Al_2O_3/Si$ | 9.5 ± 0.4 | 149 ± 3 | 10.0 ± 0.2 | 154 ± 2 | 10.7 ± 0.2 | 155 ± 2 |

According to the nanoindentation measurements, the elastic moduli of the laminates were similar and fell between the moduli of $Ta_2O_5$ and $Al_2O_3$ reference films (Table 3). The elastic modulus of the laminates did not indicate the presence of layers with different elasticities, while the difference between $Al_2O_3/Si$ (~160 GPa) and $Ta_2O_5/Si$ (~139 GPa) moduli was moderate, ~20 GPa.

Scanning probe microscopy images were taken of four indents on each sample, and an average descriptive image was chosen for each sample. Weak pile-up of 2 and 3 nm was observed on the substrate and the $Al_2O_3$ film, respectively, being only slightly higher than general surface roughness. However, $Ta_2O_5$ tended to pile up more significantly, up to 7 nm (not shown). The pile-up around the indents of the double-, triple- and quadruple-layered laminates was around 4 nm (Figure 6). Even for the $Ta_2O_5/Al_2O_3/Ta_2O_5/Si$ sample, the pile-up remained moderate, although it consisted mostly of $Ta_2O_5$, which tended to pile up most strongly. Intermediate $Al_2O_3$ layer decreased the pile-up behavior of the material.

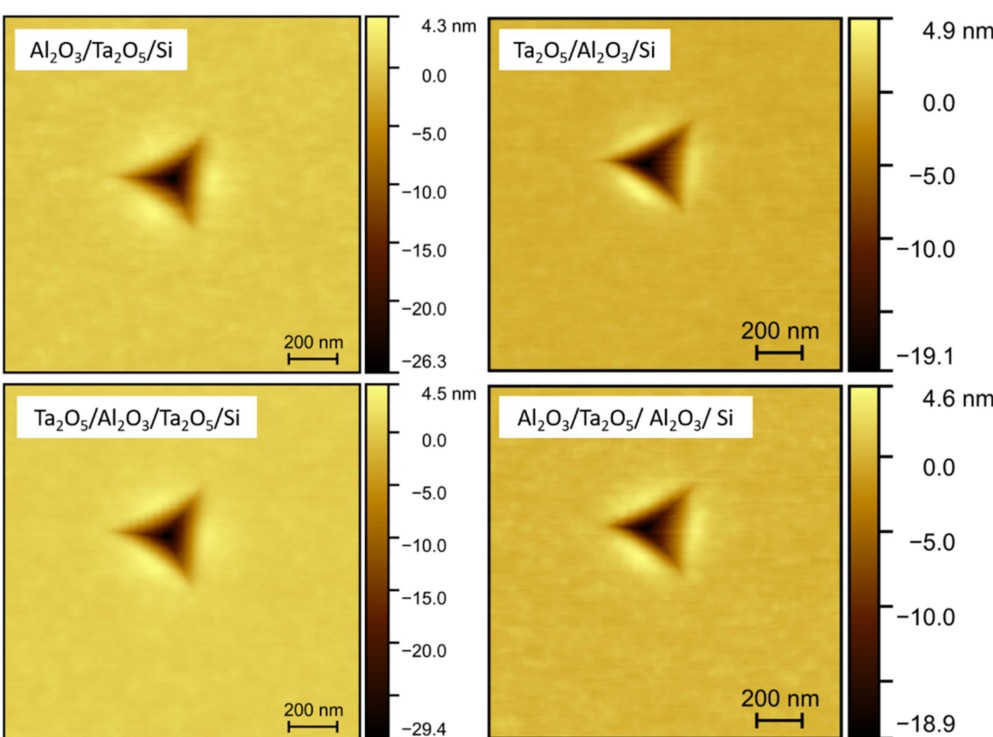

**Figure 6.** Scanning probe microscopy images of idents on double- and triple-layered laminates.

## 4. Discussion

Pelegri et al. [36] investigated theoretically the mechanical behavior of soft film-hard substrate and hard film-soft substrate systems during nanoindentation. The model materials possessed a difference in hardness within 27 GPa and resulted in hardness-displacement graphs wherein, due to the effect of the substrate, the hardness started to increase with displacement at 45% into the thickness of the soft film on hard substrate. At the same time, the hardness started to decrease only until displacement reached 30% of the thickness of the hard film on soft substrate. There was more pile-up occurring for the soft film-hard substrate system compared to the bulk soft material [36]. This approach could be used to explain the hardness-displacement curves of the double-layered laminates in the present study. The mechanical behaviour of the $Ta_2O_5/Al_2O_3/Si$ sample changed at around 20 nm displacement when the softer top $Ta_2O_5$ layer was penetrated about 60% (Figure 5a). Beginning from the displacement of about 33 nm, the hardness was slightly changed again, which is around 50% of the total laminate thickness. A displacement where the mechanical property varied could be considered where the underlying material started to influence the nanoindentation. Therefore, the hardness of 35 nm thick $Ta_2O_5$ began to be affected by the harder $Al_2O_3$ underlayer at 60% of the thickness of $Ta_2O_5$. The $Ta_2O_5$ layer decreased the overall hardness of the laminate, as the $Ta_2O_5/Al_2O_3$-Si system demonstrated a rather high difference in hardness along the indentation depth, and the substrate started to affect mechanical properties already at 50% of the thickness of the laminate. The depth where the substrate starts affecting the nanoindentation seemed to depend on the difference in the mechanical properties of the film and substrate materials.

The sample $Al_2O_3/Ta_2O_5/Si$ contained intrinsic transition from a harder material to softer material. Indentation until 30% of the top $Al_2O_3$ layer thickness, that is at 12 nm in depth, resulted in the maximum hardness value (Figure 5a). The displacement with the minimum hardness occurred at the depth of 34% of the total thickness of the laminate. The latter result indicates that, in the case of the $Al_2O_3/Ta_2O_5$-Si system, the substrate seems to affect measurement at smaller depths compared to the $Ta_2O_5/Al_2O_3/Si$ laminate, where the influence of the substrate became apparent at 50% of the thickness. Besides affecting the hardness near the surface, the sequence of the layers seemed to have some influence

on the hardness near the substrate, too. In the latter case, however, the hardness of both double-layered laminates started to overlap at higher displacements (Figure 5a) (Table 3).

$Ta_2O_5/Al_2O_3/Ta_2O_5/Si$ and $Al_2O_3/Ta_2O_5/Si$ samples behaved similarly under the nanoindentation (Figure 5b). The difference between these laminates, as engineered, was the additional softer $Ta_2O_5$ top layer and the slightly lower layer thickness, so that the similar shape of the hardness-displacement curve could be attributed to the $Al_2O_3/Ta_2O_5/Si$ structure, which also constitutes the $Ta_2O_5/Al_2O_3/Ta_2O_5/Si$ film. The width of the peak on the curve was thinner for the triple-layered laminate compared to the double-layered one. This was probably due to the lower thickness of the layers. Complementarily, the additional $Ta_2O_5$ top layer or the indenter tip radius might affect the shape of the hardness-displacement curve. Lofaj et al. [37] found, both experimentally and theoretically, that the tip radius influences the resolution of the nanoindentation measurement in the case of thin films, i.e., to correctly determine the maximum hardness of a thin hard film on softer substrate the indenter tip radius must be sufficiently sharp. Hence, for nanolaminates with thin layers, the indenter tip radius might have an influence and must be taken into consideration. In this study, however, the indenter tip radius was undetermined, but the tip was calibrated (Figure 2) and gave correct results in the displacement range of 10 to 55 nm on the fused quartz glass.

In the case of the $Al_2O_3/Ta_2O_5/Al_2O_3/Si$ sample, the mechanically different layers remained undistinguishable on the hardness-displacement graph (Figure 5b). The $Ta_2O_5$ layer between two $Al_2O_3$ might have a smaller influence on the hardness as the deformation of the softer layer is impeded by the surrounding harder material, and as for the previous laminate, the harder layer can influence the deformation of the surrounding softer material. The quadruple-layered laminate acted similarly steady under nanoindentation conditions. However, in this case, the top $Ta_2O_5$ layer was not completely surrounded by the harder material, meaning the layer thickness also has an important influence on the mechanical properties of the laminates.

## 5. Conclusions

Amorphous $Al_2O_3$-$Ta_2O_5$ nanolaminates were atomic layers deposited on Si with an approximate thickness of 70 nm. The laminates possessed hardness in the range of 8.2–11.1 GPa. The hardness of the surface of the double-and triple-layered laminates was affected by the sequence of the layers, and quite possibly, the hardness at higher depths was also somewhat affected. The thickness of layers was also significant as the peak in the hardness-displacement curve appeared slimmer and vaguer as layer thickness decreased. Triple-layered $Al_2O_3/Ta_2O_5/Al_2O_3/Si$ laminate and quadruple-layered $Ta_2O_5/Al_2O_3/Ta_2O_5/Al_2O_3/Si$ laminate possessed a steady change of hardness with similar values along the depth, although consisted of various amounts of the oxides with a 3–4 GPa difference in hardness. The elastic modulus of amorphous $Al_2O_3$-$Ta_2O_5$ nanolaminates did not depend on the layer structure and fell between 145 and 155 GPa for all the laminates. The difference of hardness of the materials in the film-substrate systems seemed to affect the depth where the substrate started to influence nanoindentation. The beneficial effects of annealing on the mechanical behavior of the $Al_2O_3$-$Ta_2O_5$ double- and triple-layered nanolaminates will be investigated in a future study.

**Author Contributions:** Conceptualization, T.J. and H.-M.P.; methodology, P.R., T.J. and H.-M.P.; software, P.R., T.J. and H.-M.P.; validation, P.R., T.J. and H.-M.P.; resources, K.K. and A.T.; writing—original draft preparation, H.-M.P.; writing—review and editing, T.J., K.K. and A.T.; funding acquisition, K.K. and A.T. All authors have read and agreed to the published version of the manuscript.

**Funding:** The present study was partially funded by the European Regional Development Fund project "Emerging orders in quantum and nanomaterials" (Grant No. TK134) and Estonian Research Agency (Grant Nos. PRG4 and PRG753).

**Institutional Review Board Statement:** Not applicable.

**Informed Consent Statement:** Not applicable.



**Data Availability Statement:** The data that support the findings of this study are available from the corresponding author upon reasonable request.

**Acknowledgments:** Acknowledgements to Aivar Tarre, Aarne Kasikov, Hugo Mändar and Lauri Aarik for assisting with atomic layer deposition and film characterization measurements. This work was partially supported by the ERDF project "Center of nanomaterials technologies and research" (NAMUR+, Project No. 2014-2020.4.01.16-0123).

**Conflicts of Interest:** The authors declare no conflict of interest.

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
