# Peer review of "Influence to Hardness of Alternating Sequence of Atomic Layer Deposited Harder Alumina and Softer Tantala Nanolaminates"

_coatings, doi:10.3390/coatings12030404_

Round 1

Reviewer 1 Report

In this paper, the authors studied the influence to hardness of alternating sequence of atomic layer deposited harder alumina and softer tantala nanolaminates. This work is written well and organized reasonably. I wonder to recommend it for publication in Coating after some revisions.

1. The film is very thin, hence, the indentation results should be discussed based some external effects, such as size effect (International Journal of Plasticity, 2012, 34: 1-11), substrate effect (Journal of Materials Science & Technology, 2012, 28 (7): 626-635), creep effect (Journal of Materials Science, 2008, 43 (17): 5952-5955), ect.

2. The authors should provide the indentation conditions, including loading time, unloading time. It is better if the authors can add some typical load-displacement curves.

3. This work present that "The sequence of the oxides from surface to substrate along with the layer thickness had an influence on the hardness causing rises and declines in hardness along the depth, yet did not affect the elastic modulus." More physical mechanism should be discussed.

Reviewer 2 Report

In the manuscript, the authors studied the nanomechanical properties of ALD deposited multi-layered thin films using nanoindentation. The authors studied several double-layered and triple-layered thin films with alternating harder and softer compounds, together with reference materials. Generally, the paper was well written. The authors have provided sufficient background information, described their experimental design in detail, and supported their conclusions with sufficient data. I do not see significant flaws in the manuscript.

  1. Starting from Line 95, the authors stated that their nanoindentation was carried out in “continuous stiffness measurement mode which obtained several tens of datapoints over a displacement range for a single indentation”. Can they control the exact displacement for one data point? I am asking because the authors provided hardness and module data at 10 nm, 25 nm, and 40 nm. Are those data taken at the exact depth?
  2. Just out of curious, have the authors considered using cubic corner tip for their measurement? From my experience, the cubic corner tip generally gives a different hardness value (more reliable if calibrated well) for small indents, and the data gradually overlaps with the berkovich tip at higher depth. However, I do not believe the authors need to provide additional data. They have already calibrated their berkovich tip well and their data is valid and already sufficient enough.

Reviewer 3 Report

The article deals with the extremely interesting and timely topic of determining the mechanical properties of nanolayers produced by the ALD method. The article was written correctly. It is legible and its design is thoughtful and legible. All drawings are of good quality and well prepared. The article is written correctly in terms of style and English. Discussion with a summary and conclusions described broadly and correctly. Up-to-date literature. I propose to print the article without any changes and assign it to SI: Recent Advance in Thin Films Deposited by Vacuum Methods for Optics, Electronics and Medicine

Author Response

Thank You to the reviewer for the evaluation and the comments.